# Structure of the Lifeact–F-actin complex

**Alexander Belyy**, **Felipe Merino**¤, **Oleg Sitsel**, **Stefan Raunser***

Department of Structural Biochemistry, Max Planck Institute of Molecular Physiology, Dortmund, Germany

¤ Current address: Department of Protein Evolution, Max Planck Institute for Developmental Biology, Tübingen, Germany

* stefan.raunser@mpi-dortmund.mpg.de

## Abstract

Lifeact is a short actin-binding peptide that is used to visualize filamentous actin (F-actin) structures in live eukaryotic cells using fluorescence microscopy. However, this popular probe has been shown to alter cellular morphology by affecting the structure of the cytoskeleton. The molecular basis for such artefacts is poorly understood. Here, we determined the high-resolution structure of the Lifeact–F-actin complex using electron cryo-microscopy (cryo-EM). The structure reveals that Lifeact interacts with a hydrophobic binding pocket on F-actin and stretches over 2 adjacent actin subunits, stabilizing the DNase I-binding loop (D-loop) of actin in the closed conformation. Interestingly, the hydrophobic binding site is also used by actin-binding proteins, such as cofilin and myosin and actin-binding toxins, such as the hypervariable region of TccC3 (TccC3HVR) from *Photorhabdus luminescens* and ExoY from *Pseudomonas aeruginosa*. In vitro binding assays and activity measurements demonstrate that Lifeact indeed competes with these proteins, providing an explanation for the altering effects of Lifeact on cell morphology in vivo. Finally, we demonstrate that the affinity of Lifeact to F-actin can be increased by introducing mutations into the peptide, laying the foundation for designing improved actin probes for live cell imaging.

**Data Availability Statement:** The coordinates for the cryo-EM structure of F-actin-Lifeact have been deposited in the Electron Microscopy Data Bank under accession number EMD-11721. The corresponding molecular model has been

## Introduction

The network of actin filaments in eukaryotic cells is involved in processes ranging from intracellular trafficking to cell movement, cell division, and shape control [1]. It is therefore not surprising that much effort has been directed to characterize the actin cytoskeleton under both physiological and pathological conditions. Numerous actin-visualizing compounds were developed to enable this. These include small molecules, labeled toxins, recombinant tags, as well as actin-binding proteins and peptides (see [2] for a detailed review). However, using these molecules to study actin in vivo often alters the properties of actin filaments to such an extent that normal homeostasis of the cytoskeleton is impaired. Since these side effects cannot be avoided, it is important to know their molecular basis in order to be able to adequately interpret the experimental data.

Phalloidin and jasplakinolide are cyclic peptides derived from the death cap mushroom *Amanita phalloides* and marine sponge *Jaspis johnstoni* [3,4], respectively. They bind specifically to filamentous actin (F-actin), and when fused to a fluorescent probe, their derivatives allow visualization of the cell cytoskeleton by fluorescence microscopy [2]. However, both

deposited at the wwPDB with accession code PDB 7AD9.

**Funding:** This work has been funded by the Max Planck Society (S.R.). A.B. is a fellow of the Humboldt foundation. The funders had no role in study design, data collection and analysis, decision to publish, or preparation of the manuscript.

**Competing interests:** The authors have declared that no competing interests exist.

**Abbreviations:** cAMP, 3′,5′-cyclic adenosine monophosphate; cGMP, 3′,5′-cyclic guanosine monophosphate; cryo-EM, electron cryo-microscopy; D-loop, DNase I–binding loop; F-actin, filamentous actin; GFP, green fluorescent protein; HEK, human embryonic kidney; LB, lysogeny broth; MBP, maltose-binding protein; NM2C, non-muscle myosin 2C; PVDF, polyvinylidene difluoride; SD2, subdomain 2; TccC3, TccC3HVR, hypervariable region of TccC3; WT, wild-type; YPD, yeast extract peptone.

molecules strongly stabilize F-actin and shift the cellular actin equilibrium, largely limiting their use in live cell imaging [5,6]. Recent electron cryo-microscopy (cryo-EM) studies from our group and others have uncovered how phalloidin and jasplakinolide affect the structure of F-actin and described the potential limitations of their use [7–9].

The development of various fluorescent proteins provided new ways of visualizing the actin cytoskeleton. A simple and popular technique compatible with live cell imaging is to express actin fused to green fluorescent protein (GFP)-like proteins [10]. However, such actin chimeras often interfere with the normal functionality of the cytoskeleton in a way that results in experimental artefacts [11,12]. An alternative to GFP–actin is to fuse GFP to actin-binding proteins, such as utrophin [13,14] or *Arabidopsis* fimbrin [15], and synthetic affimers that bind actin [16,17]. These actin filament markers have been successfully used in a variety of cell types and organisms [2,18,19].

The most recent development is an F-actin-binding nanobody called Actin-Chromobody that claims to have a minimal effect on actin dynamics and no notable effect on cell viability [20]. However, the binding of the Actin-Chromobody to actin has not yet been characterized at molecular level, leaving the true extent of possible side effects open.

As described above, fluorophore-bound proteins are large and bulky, resulting in possible steric clashes when interacting with actin. To avoid these problems, small fluorophore-labeled peptides were developed. The most commonly used one is Lifeact, which is a 17-amino acid peptide derived from the N-terminus of the yeast actin-binding protein ABP140. In the original publication, Lifeact was described as a novel F-actin probe that does not interfere with actin dynamics in vitro and in vivo [21]. The same group later reported transgenic Lifeact–GFP-expressing mice that were phenotypically normal and fertile [22], and no influence of Lifeact on cellular processes was found under the published experimental conditions. Two other groups performed a direct comparison of various F-actin-binding probes and confirmed the low influence of Lifeact on cell cytoskeletal architecture [23,24]. Later on, however, several major Lifeact-caused artefacts were described: Lifeact was unable to stain certain F-actin-rich structures [25,26], it disturbed actin assembly in fission yeast [27], it caused infertility and severe actin defects in *Drosophila* [19], and it altered cell morphology in mammalian cells [28]. The existing explanatory hypothesis suggests that Lifeact induces a conformational change in F-actin that affects binding of cofilin and eventually impairs cell cytoskeletal dynamics [27]. However, despite the widespread usage of Lifeact, the validity of this hypothesis is still a matter of debate since no structure of Lifeact-decorated F-actin has been available.

In order to address this, we solved the structure of the Lifeact–F-actin complex using cryo-EM. The 3.5 Å structure reveals that Lifeact binds to 2 consecutive actin subunits of the same strand of the filament and displaces the DNase I–binding loop (D-loop) upon binding. The binding site overlaps with that of cofilin and myosin, suggesting that artefacts in live cell imaging are caused by competition between these proteins and Lifeact. Competition binding assays in vitro prove that this is indeed the case. Furthermore, we show that the binding of Lifeact to F-actin considerably reduces the in vivo toxicity of the actin-modifying hypervariable region of TccC3 (TccC3HVR), a toxin from *Photorhabdus luminescens*. Our data will help to predict potential artefacts in experiments using Lifeact and will serve as a strong basis for developing new actin-binding probes with improved properties.

## Materials and methods

### Plasmids, bacteria and yeast strains, growth conditions

The complete list of used oligonucleotides, constructs, and strains can be found in S3 Table. *Escherichia coli* strains were grown in lysogeny broth (LB) medium supplemented with

ampicillin (100 μg/ml) or kanamycin (50 μg/ml). *S. cerevisiae* were grown on rich yeast extract peptone (YPD) medium or on synthetic defined medium (Yeast nitrogen base, BD, United States of America) containing galactose or glucose and supplemented if required with uracil, histidine, leucine, tryptophan, or adenine. *S. cerevisiae* strains were transformed using the lithium acetate method [37]. Yeast actin mutagenesis was performed as described previously [38]. Yeast viability upon Lifeact–maltose-binding protein (MBP) overexpression under the galactose promoter was analyzed by a drop test: 5-fold serial dilutions of cell suspensions were prepared from overnight agar cultures by normalizing $OD_{600}$ measurements, then spotted onto agar plates, and incubated for 2 to 3 days at 30°C. Analysis of protein expression in yeast was performed following the described protocol [39]: Yeast cells were grown in liquid galactose–containing medium overnight at 30°C. Cells corresponding to 1 ml of $OD_{600}$ 1.0 were washed with 0.1 M NaOH, resuspended in 50 μl of 4-fold Laemmli sample buffer, and boiled for 5 minutes at 95°C. 5 μl of the extracts were separated by SDS-PAGE, followed by Western blotting analysis, and incubation with anti-MBP (NEB, USA) or anti-RPS9 serum (polyclonal rabbit antibodies were a generous gift of Prof. S. Rospert). Uncropped gels and Western blots are presented in S4 Fig.

## Protein expression and purification

*P. aeruginos*a ExoY toxin fused to MBP and Lifeact variants fused to mCherry were purified from *E. coli* BL21-CodonPlus(DE3)-RIPL cells harboring the plasmids 2479, 2556, 2557, or 2558. A single colony was inoculated in 100 ml of LB media and grown at 37°C. At $OD_{600}$ 1.0, protein expression was induced by the addition of IPTG to a final concentration of 1 mM. After 2 hours for ExoY-MBP or 16 hours for Lifeact–mCherry of expression at 37°C, the cells were harvested by centrifugation, resuspended in buffer A (20 mM Tris pH 8 and 500 mM NaCl), and lysed by sonication. The soluble fraction was applied on buffer A-equilibrated Protino Ni-IDA resin (Macherey-Nagel, Germany), washed, and eluted by buffer A, supplemented with 250 mM imidazole. Finally, the eluates were dialyzed against buffer B (20 mM Tris pH 8 and 150 mM NaCl) and stored at −20°C.

Rabbit skeletal muscle α-actin was purified as described previously [8] and stored in small aliquots at −80°C.

Human cofilin-1 was purified from *E. coli* cells using previously described method [40]. In short, Rosetta DE3 *E. coli* cells were transformed with the 1855 cofilin plasmid. An overnight culture derived from a single colony was diluted into 2 L of LB media to $OD_{600}$ 0.06 and grown at 37°C. When $OD_{600}$ reached 0.7, the cells were cooled to 30°C, and cofilin expression was induced by adding IPTG to a final concentration of 0.5 mM. After 4 hours of expression, the cells were harvested by centrifugation, resuspended in buffer C (10 mM Tris pH 7.8, 1 mM EDTA, 1 mM PMSF, and 1 mM DTT), and lysed using a fluidizer. The soluble fraction of the lysate was dialyzed overnight in buffer D (10 mM Tris pH 7.8, 50 mM NaCl, 0.2 mM EDTA, and 2 mM DTT) and cleared by centrifugation. Then, the lysate was applied onto DEAE resin and washed with buffer D. Cofilin-containing fractions of the flow through were collected and dialyzed against buffer E (10 mM PIPES pH 6.5, 15 mM NaCl, 2 mM DTT, and 0.2 mM EDTA). After centrifugation, the protein was loaded onto Mono S cation exchange column and eluted by a linear gradient of 15 mM to 1 M NaCl in buffer E. Cofilin-containing fractions were concentrated to 10 mg/ml and stored at −80°C.

Human tropomyosin was purified from *E. coli* BL21(DE3) cells transformed with a 1609 pET19 tropomyosin plasmid using the previously described method [41] with minor modifications. In brief, an overnight culture derived from a single colony was diluted into 5 L of LB media to $OD_{600}$ 0.06 and grown at 37°C. When the $OD_{600}$ reached 0.5, the cells were cooled

down to 20˚C and recombinant protein expression was induced by adding IPTG to a final concentration of 0.4 mM. After overnight protein expression, the cells were harvested by centrifugation, resuspended in buffer H (20 mM Tris pH 7.5, 100 mM NaCl, 5 mM MgCl$_2$, 2 mM EGTA, and Roche cOmplete protease inhibitor (Sigma-Aldrich, USA)), and lysed by fluidizer. The soluble fraction of the lysate was heated for 10 minutes at 80˚C, then cooled down to 4˚C and centrifuged. The supernatant was mixed 1:1 with buffer I (20 mM sodium acetate buffer pH 4.5, 100 mM NaCl, 5 mM MgCl$_2$, and 2 mM EGTA). The precipitate was collected and incubated for 1 hour with buffer J (10 mM Bis-Tris pH 7 and 100 mM NaCl). The renatured protein was applied to a HiTrap Q anion exchange column (Sigma-Aldrich) and eluted by a linear gradient of 100 mM to 1 M NaCl in buffer J. Tropomyosin-containing fractions were pooled and stored at −80˚C.

The motor domain of nonmuscular myosin-2C (MYH14, isoform 2 from *Homo sapiens*) consisting of amino acids 1–799 was purified as described previously [32]

Tcc3HVR, the ADP-ribosyltransferase domain of the *P. luminescens* TccC3 protein (amino acids 679–960), was purified as described previously [42].

The TcdA1 and TcdB1-TccC3 components of the *P. luminescens* PTC3 toxin were expressed and purified as described previously [43].

## Cryo-EM sample preparation, data acquisition, and processing

Actin was polymerized by incubation in F-buffer (120 mM KCl, 20 mM Tris pH 8, 2mM MgCl2, 1 mM DTT, and 1 mM ATP) in the presence of a 2-fold molar excess of phalloidin for 30 minutes at room temperature and further overnight at 4˚C. The next day, actin filaments were pelleted using a TLA-55 (Beckman Coulter, USA) rotor for 30 minutes at 150,000 g at 4˚C and resuspended in F-buffer. 5 minutes before plunging, F-actin was diluted to 6 μM and mixed with 200-μM Lifeact peptide (the peptide of sequence MGVADLIKKAESISKEE with carboxyl-terminal amide modification was provided by Genosphere, France with >95% purity). The peptide was dissolved in 10 mM Tris pH 8. To improve ice quality, Tween-20 (Calbiochem, USA) was added to the sample to a final concentration of 0.02% (w/v). Plunging was performed using the Vitrobot Mark IV system (Thermo Fisher Scientific, USA) at 13˚C and 100% humidity: 3 μl of sample were applied onto a freshly glow-discharged copper R2/1 300 mesh grid (Quantifoil, Germany), blotted for 8 seconds on both sides with blotting force −20 and plunge-frozen in liquid ethane.

The dataset was collected using a Talos Arctica transmission electron microscope (Thermo Fisher Scientific) equipped with an XFEG at 200 kV using the automated data collection software EPU (Thermo Fisher Scientific). Two images per hole with defocus range of −0.6 to −3.35 μm were collected with the Falcon III detector (Thermo Fisher Scientific) operated in linear mode. Image stacks with 40 frames were collected with total exposure time of 3 seconds and total dose of 60 e⁻/Å. A total of 1,415 images were acquired, and 915 of them were further processed. Filaments were automatically selected using crYOLO (Max-Planck-Society e.V., Germany) [44]. Classification, refinement, and local resolution estimation were performed in SPHIRE (Max-Planck-Society e.V.) [45]. Erroneous picks were removed after a round of 2D classification with SPHIRE. After removing bad particles, further segments were removed to ensure that each filament consists of at least 5 members. After 3D refinement, particles were polished using Bayesian polishing routine in Relion [46] and refined once again within SPHIRE.

## Model building and design

We used the structure of actin in complex with ADP–P$_i$ (PDBID 6FHL) as starting model for the filament and built a de novo model of the Lifeact peptide using Rosetta's fragment-based

approach [47]. Since 1 residue of Lifeact is missing from the density, we threaded the sequence in all possible registers—including the reverse orientations—and minimized them in Rosetta. The solution starting from M1 with the N-terminus pointing toward the pointed end was clearly better than all the others (S2 Fig). A set of restraints for phalloidin was built with eLBOW (Lawrence Berkeley National Laboratory, USA) [48], and the toxin was manually fit into the density with Coot (University of Oxford, England) [49]. The model was further refined using iterative rounds of Rosetta's fragment-based iterative refinement [50] and manual building with Coot and ISOLDE (University of Cambridge, England) [49,51]. The model was finally refined within Phenix (University of California, USA) [52] to fit B-factors and correct the remaining geometry errors. Cryo-EM data collection, refinement, and validation statistics are available in S1 Table.

We used RosettaScripts [30] to design a version of Lifeact with improved affinity. The input protocol and starting files are available in the Supporting information (S1 File). The figures were made using UCSF Chimera (University of California, USA) [53].

## Yeast confocal microscopy

Yeast cells bearing plasmids that encode Lifeact–mCherry variants under the native ABP140 promoter and terminator were grown overnight on a liquid synthetic defined medium (Yeast nitrogen base, BD, USA) supplemented with glucose. On the following day, the cultures were diluted to $OD_{600}$ 0.5 using fresh media, incubated for 2 to 3 hours at 30°C and centrifuged at 6,000 g. The cell pellet was washed twice with PBS buffer, fixed by 4% formaldehyde for 20 minutes at room temperature, washed once with PBS buffer, and stained with the ActinGreen 488 probe (Invitrogen, USA). After 30 minutes incubation with the probe at room temperature, the cells were washed twice in PBS and applied to concanavalin A-coated glass bottom Petri dishes. Image acquisition was performed with a Zeiss LSM 800 confocal laser scanning microscope (Zeiss, Germany), equipped with 63× 1.4 DIC III M27 oil immersion objective and 2 lasers with wavelengths of 488 and 561 nm. The yeast images presented on Fig 2 were smoothened with a Gaussian filter. The fluorophore signal strength on the presented micrographs was not normalized. To avoid experimental bias, we measured the weighted colocalization coefficient of phalloidin with Lifeact–mCherry in 15 cells of each strain using ZEN software (Zeiss). To calculate the ratio of Lifeact–mCherry to phalloidin fluorescence signal in patches (Fig 3D), we manually picked F-actin-rich patches and measured the total fluorescence signals of Lifeact–mCherry and phalloidin in the 1-$\mu m^2$ square regions. In total, 15 measurements with the similar phalloidin intensity from every yeast strain were used to calculate the ratio that were plotted onto the graph. To estimate the signal-to-noise ratio of the Lifeact variants (Fig 3E), we firstly measured the total fluorescence signal of Lifeact–mCherry in the 0.25-$\mu m^2$ square regions in the middle of the cell where no phalloidin F-actin staining was present ("background value"). Then, we measured the total fluorescence signals of Lifeact–mCherry and phalloidin in the 0.25-$\mu m^2$ square regions consisting of an F-actin-rich patch. In total, 15 measurements of Lifeact–mCherry signal with the similar phalloidin intensities from every yeast strain were taken into account, divided by the background value of Lifeact–mCherry signal of the corresponding cell, and plotted onto the graph.

## Cosedimentation assays

F-actin was prepared as follows. An aliquot of freshly thawed G-actin was centrifuged at 150,000 g using a TLA-55 rotor for 30 minutes at 4°C to remove possible aggregates. Then, actin was polymerized by incubation in F-buffer (120 mM KCl, 20 mM Tris pH 8, 2 mM $MgCl_2$, 1 mM DTT, and 1 mM ATP) for 2 hours at room temperature or overnight at 4°C. For

the cosedimentation experiments with myosin and tropomyosin, a 2-fold molar excess of phalloidin was added after the polymerization. Then, the actin filaments were pelleted using a TLA-55 rotor at 150,000 g for 30 minutes and resuspended in the following buffers: F-buffer was used for cosedimentations with tropomyosin or Lifeact–mCherry fusion proteins; 20 mM HEPES pH 6.5, 50 mM KCl, and 2 mM $MgCl_2$ was used for cosedimentations with cofilin; and 120 mM KCl, 20 mM Tris pH 8, 2 mM $MgCl_2$, and 1 mM DTT was used for cosedimentations with myosin.

Cosedimentation assays were performed in 30-μl volumes by first incubating F-actin with the specified proteins for 5 minutes at room temperature, then centrifuging at 120,000 g using the TLA120.1 rotor for 30 minutes at 4˚C. For the competition assays, Lifeact peptide was added to the mixture at the specified concentrations. After centrifugation, aliquots of the supernatant and resuspended pellet fractions were separated by SDS-PAGE using 4% to 15% gradient TGX gels (Bio-Rad, USA) and analyzed by densitometry using Image Lab software (Bio-Rad). The $K_d$ values were calculated as previously described using GraphPad Prism software (GraphPad Software, USA) [34].

## ADP-ribosylation of actin by TccC3HVR

A total of 8-μl mixtures of 1 μg (2.4 μM) actin and Lifeact peptide at specified concentrations were preincubated for 5 minutes at room temperature in TccC3 buffer (1 mM NAD, 20 mM Tris pH 8, 150 mM NaCl, and 1 mM $MgCl_2$). The ADP-ribosylation reaction was initiated by addition of 0.02 μg (61 pM) of TccC3HVR into the mixture. After 10 minutes of incubation at 37˚C, the reaction was stopped by adding Laemmli sample buffer (62.5 mM Tris-HCl pH 8.0, 25 mM DTT, 1.5% SDS, 10% glycerol, and 0.1 mg/ml bromphenol blue) and heating the sample at 95˚C for 5 minutes. Components of the mixture were separated by SDS-PAGE, blotted onto a polyvinylidene difluoride (PVDF) membrane using a Trans-Blot Turbo Transfer System (Bio-Rad), and visualized using a combination of anti-mono-ADP-ribose binding reagent (Merck, Germany) and anti-rabbit-HRP antibody (Bio-Rad). The level of actin ADP-ribosylation was quantified by densitometry using Image Lab software (Bio-Rad).

## Confocal microscopy of mammalian cells and the in vivo intoxication competition assay

$0.05 \times 10^6$ human embryonic kidney (HEK) 293T cells were seeded in a 35-mm glass-bottom, poly-L-lysine coated Petri dishes in 2-mL DMEM/F12 + 10% FBS media and grown for 24 hours in a 5% $CO_2$ atmosphere at 37˚C. The cells were then transfected with mCherry fusions of actin or Lifeact variants using the FuGENE transfection reagent (Promega, USA). The cells were grown for a further 24 hours and transferred to an LSM 800 microscope (Zeiss) equipped with a C-Apochromat 40×/1.2 W objective and maintained in a 5% $CO_2$ atmosphere at 37˚C. Images were acquired using the Airyscan detector (Zeiss) and a 561-nm laser wavelength for excitation. After taking the first image (0 hour), 300 pM PTC3 was added (preformed by mixing purified TcdA1 and TcdB2-TccC3 in a 5:1 molar ratio). Images were taken at 10-minute intervals for 5 hours.

The images were processed in Fiji [54]. The cells were selected by applying a threshold on the red fluorescence channel, then their areas were measured, normalized to the original cell area at the 0-hour time point, and the resulting % change in cell area that occurred during the experiment was plotted.

Three independent experiments for each construct were performed. The cells were not tested for *Mycoplasma* contamination.

## Results and discussion

### Structure of the Lifeact–F-actin complex

Prior studies have indicated a molecular basis for stabilization of actin filaments by phalloidin [7–9]. When phalloidin is added during actin polymerization, it maintains the ADP–$P_i$ state of actin and stabilizes the D-loop in its open conformation [7], making it a suitable reference state for structural studies. We therefore polymerized actin in the presence of phalloidin and added an excess of Lifeact to the formed filaments in order to fully decorate the filaments with Lifeact. We then determined the structure of this complex by cryo-EM (Fig 1, S1 Table). The average resolution of the reconstruction was 3.5 Å, with local areas reaching 3.0 Å (S1 Fig), which allowed us to build an atomic model in which we could reliably position most of the side chains. We could clearly identify densities corresponding to ADP, $Mg^{2+}$, and $P_i$ in the nucleotide binding pocket of actin (Fig 1A) and a density corresponding to phalloidin at the expected position [7–9] in the center of the filament (Fig 1A). Lifeact was well resolved, and we could unambiguously fit 16 out of 17 amino acids into the density (Fig 1, S2 Fig).

The peptide folds as an α-helix and spans 2 consecutive actin subunits of the same strand of the filament (Fig 1A and 1B). The binding pocket is formed by the tip of the D-loop (subdomain 2 (SD2)) of the lower subunit (M47) and SD1 of the upper subunit, where the N-terminal region of Lifeact is almost locked in by the protruding D25 of actin (Fig 1B). Although Lifeact is in general a hydrophilic peptide, it contains a hydrophobic patch formed by the side chains of V3, L6, I7, F10, and I13, which all orient to one side. This hydrophobic patch interacts with a hydrophobic groove on the surface of F-actin, which comprises M44, M47, Y143, I345, and

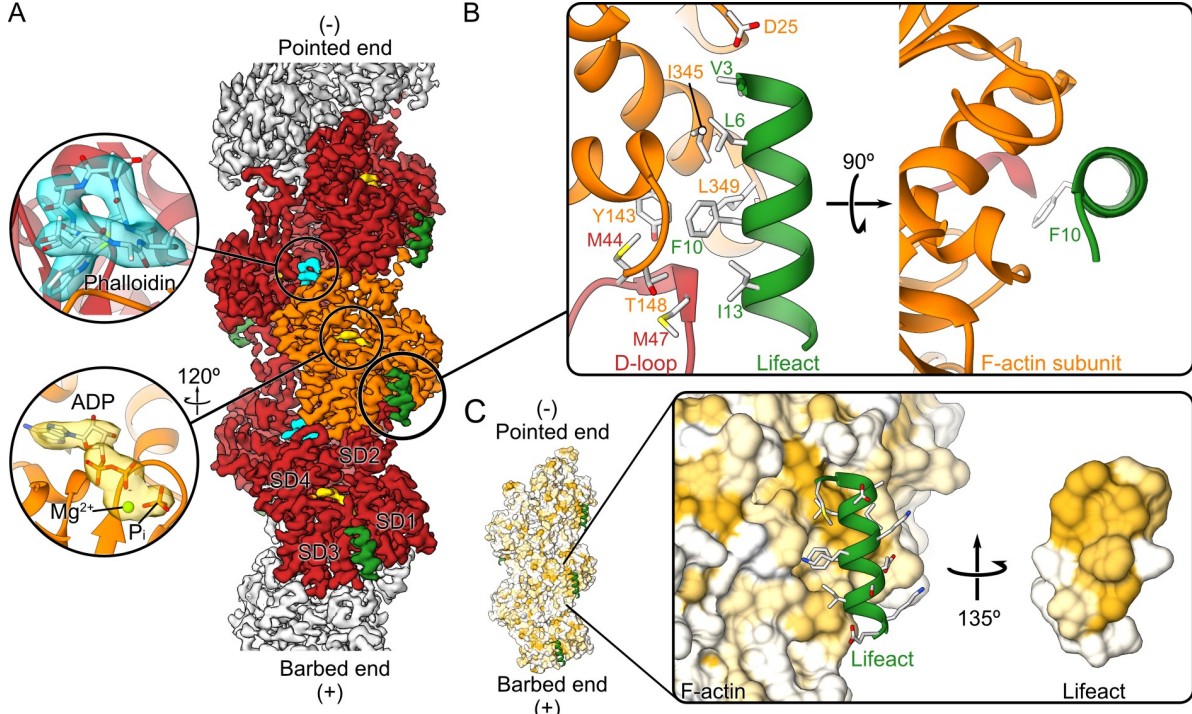

**Fig 1. Cryo-EM structure of the Lifeact–F-actin–ADP–$P_i$–phalloidin complex.** (A) The 3.5 Å resolution map of the Lifeact–F-actin–ADP–$P_i$–phalloidin complex shows a defined density for phalloidin (cyan), ADP–$P_i$ (gold), and the Lifeact peptide (green). The central subunit of actin is colored in orange, while its surrounding 4 neighbors are shown in red. (B) Atomic model of the interface between Lifeact and F-actin. (C) Surface of the atomic model of F-actin colored according to its hydrophobicity. Hydrophobicity increases as the color scale goes from white to gold. The inset highlights the hydrophobic nature of the Lifeact-binding surface. Cryo-EM, electron cryo-microscopy; D-loop, DNase I–binding loop; F-actin, filamentous actin; SD, subdomain.

L349. F10 of Lifeact is deeply buried in this pocket (Fig 1C). Interestingly and contrary to what we have seen before in samples copolymerized with phalloidin [7], the D-loop is in its closed conformation (S3A Fig). A comparison between the Lifeact–F-actin–ADP–$P_i$–phalloidin structure with that of the phalloidin-stabilized F-actin–ADP–$P_i$ [7] shows that direct interactions between Lifeact and the D-loop of F-actin are only possible if the D-loop is in its closed conformation (S1 Movie). Specifically, I13 of Lifeact interacts with M47 of F-actin, stabilizing the closed D-loop conformation in F-actin. This suggests that Lifeact has a higher affinity to F-actin–ADP, where the D-loop is in the closed conformation, than to phalloidin-stabilized F-actin–ADP–$P_i$, where the D-loop has to be first moved from the open to the closed conformation. Indeed, Kumari and colleagues [29] showed in a complementary study that the affinity of Lifeact is 3 to 4 times higher for F-actin–ADP compared to F-actin–ADP–$P_i$.

## Properties of the interaction site

Guided by the insights gained from our structure, we mutated different residues at the peptide–actin interface to study the binding properties of Lifeact in more detail. We chose to use *Saccharomyces cerevisiae* for these studies since actin mutagenesis can be easily and rapidly performed in this organism. To avoid toxicity from artificial overexpression of Lifeact, we expressed Lifeact–mCherry under the promoter of the actin-binding protein ABP140 from which Lifeact was originally derived and then performed confocal microscopy experiments to visualize Lifeact–actin interaction.

When expressing wild-type (WT) Lifeact–mCherry, we observed the typical actin-rich assemblies such as patches, cables, and contractile rings. These structures can also be observed when the cells are stained by fluorescently labeled phalloidin (Fig 2A). However, when L6 was mutated to lysine, or F10 to alanine, we did not observe these features, and Lifeact–mCherry was homogeneously distributed in the cells. Although the I13A variant displayed some of the patches, they were significantly less abundant than with WT Lifeact–mCherry (Fig 2A and 2B; data points are available in S2 Table). While the Lifeact L6K mutant introduces a charge in the hydrophobic patch, Lifeact mutants F10A and I13A reduce the hydrophobicity and change the surface structure of the peptide. Since all mutations impaired the interaction between actin and Lifeact, we conclude that hydrophobicity as well as shape complementarity are important for efficient Lifeact binding to F-actin.

Our structure suggests that actin D25 acts as an N-terminal cap for the helix of Lifeact, and the mutation D25Y would affect this interaction. In addition, mutating L349 of actin to methionine would impair its crucial interaction with Lifeact F10. Indeed, actin-rich structures were also absent when Lifeact–mCherry was expressed in cells where the endogenous actin had been replaced with the D25Y or the L349M actin variants (Fig 2C and 2D), indicating that actin D25 and Lifeact F10 are important for Lifeact binding.

To study the effect of Lifeact WT and variants on yeast viability, we overexpressed Lifeact–MBP fusions under a strong galactose promoter and analyzed their toxicity in a yeast growth phenotype assay. Consistent with a previously reported study [27], we observed that the overexpression of Lifeact–MBP caused cell toxicity (Fig 2E; uncropped images are available in S4 Fig). However, mutagenesis of I13 to alanine improved, and L6 to lysine and F10 to alanine fully restored yeast growth. Altogether, these results demonstrate the importance of shape complementarity as well as hydrophobicity at the Lifeact–actin interface.

## Lifeact mutations increase its affinity to F-actin

Despite these specific interactions between Lifeact and F-actin, the peptide binds to F-actin only with micromolar affinity [21]. A higher affinity to F-actin, together with reduced binding

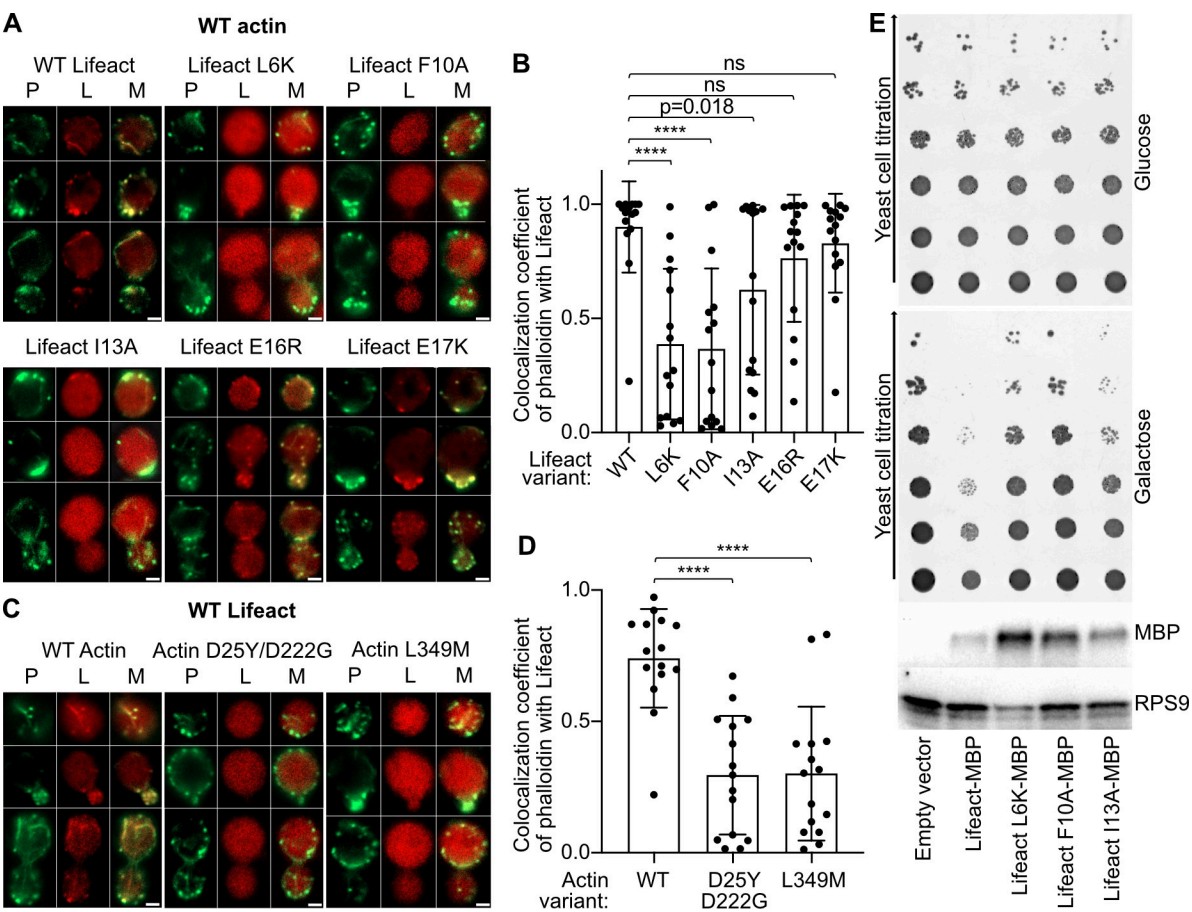

**Fig 2. The Lifeact–F-actin complex is affected by point mutations.** (A, C) Confocal microscopy images of yeast cells expressing Lifeact–mCherry variants in a WT actin background (A) and WT Lifeact–mCherry in yeast with different actin variants replacing endogenous WT actin (C). Actin was additionally stained with fluorescently labeled phalloidin (ActinGreen 488). Yeast cells at different division phases are shown. Note that for our experiments, we used the previously described D25Y/D222G double mutant of yeast actin [34]. However, D222 is located in subdomain IV and therefore unlikely plays a role in the Lifeact–F-actin interaction. Scale bars, 2 μm. (B, D) Calculated weighted colocalization coefficients of phalloidin with Lifeact–mCherry from 15 yeast cells from 2 independent experiments with 5 micrographs each, corresponding to (A) and (C), respectively. For statistical analysis, the unpaired $t$ test was used in (B) and (D). ****$p < 0.0001$; ns, not significant. The error bars in the panels correspond to standard deviations. (E) Growth phenotype assay with yeast overexpressing Lifeact–MBP variants under a strong galactose promoter. The top image marked "Glucose" corresponds to experimental conditions with low Lifeact expression. The central image marked "Galactose" corresponds to experimental conditions with high Lifeact expression. The lower image is a western blot of cells grown on galactose-containing media performed using anti-MBP and anti-Ribosomal protein S9 (RPS9) antibodies. The uncropped drop tests, western blots, and gels can be found in S4 Fig. Data points that were used to create graphs are reported in S2 Table. F-actin, filamentous actin; L, Lifeact; M, merge; MBP, maltose-binding protein; P, phalloidin; WT, wild-type.

to globular actin (G-actin), would provide a stronger signal-to-noise ratio, decreasing the background during live imaging and allowing lower expression levels of the peptide to be used during such experiments. As a proof of concept that the affinity of the Lifeact peptide can be modulated based on our Lifeact–F-actin atomic model, we attempted to increase the affinity of Lifeact to F-actin by structure-guided in silico design using RosettaScripts (https://www.rosettacommons.org/; University of Washington, USA) [30]. The design suggested several possible mutations after residue 12 of Lifeact (Fig 3A, left panel), of which E16R was especially promising. This mutation should add an additional interaction with the D-loop and an electrostatic interaction with E167 of actin (Fig 3A, right panel). To study the effects of this mutation in vitro, we purified the designed Lifeact variant fused to mCherry tag and compared its F-actin binding to WT Lifeact (Fig 3B and 3C). The measured $K_d$ between WT Lifeact–mCherry

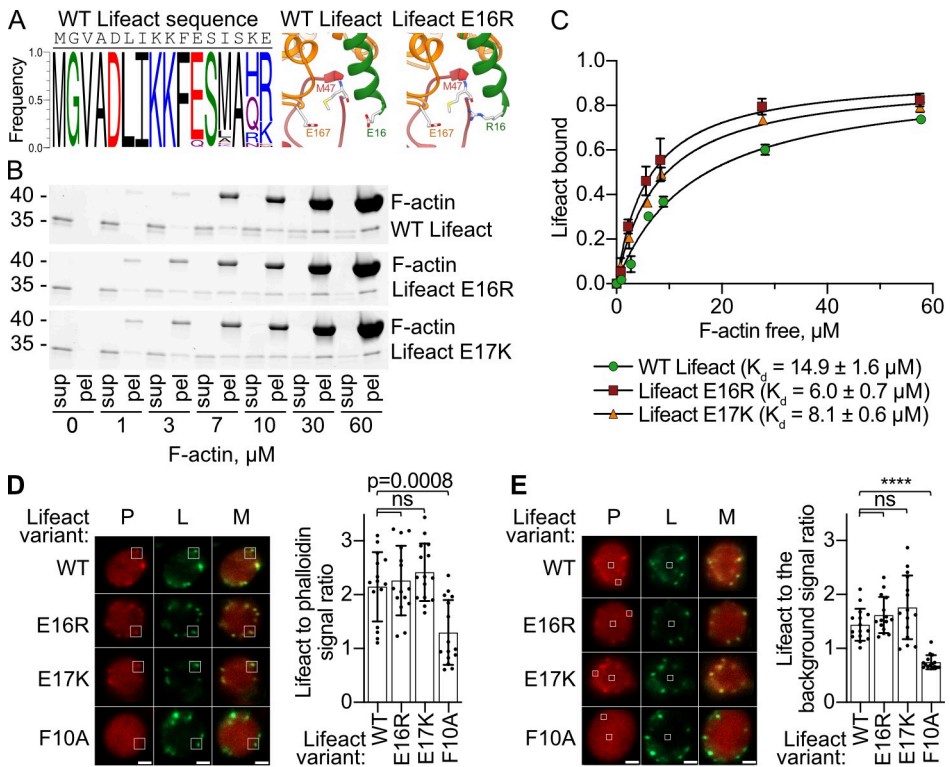

**Fig 3. Lifeact sequence design.** (A) Frequency of amino acids in the top 100 designs produced by Rosetta (left panel) and predicted structure of the E16R mutant (right panel). The WT structure is included for comparison. (B) Cosedimentation of F-actin and 3- μM Lifeact–mCherry proteins detected by SDS-PAGE. The upper band corresponds to Lifeact–mCherry, and the lower band corresponds to actin. Representative stain-free gels are shown. The uncropped gels can be found in S4 Fig. (C) The fractions of Lifeact–mCherry that cosedimented with F-actin were quantified by densitometry and plotted versus actin concentrations. The error bars in panel C correspond to standard deviations of 3 independent experiments. (D) Representative images of cells expressing the indicated Lifeact variants are shown on the left panel. The total fluorescence signal of Lifeact–mCherry from the 1- $μm^2$ square region consisting of an F-actin-rich patch was divided by the total fluorescence signal of phalloidin in the same region and plotted on the right panel. (E) The total fluorescence signal of Lifeact–mCherry from the 0.25- $μm^2$ square region (as shown on the left panel) consisting of an F-actin-rich patch was divided by the total fluorescence signal of Lifeact–mCherry in the F-actin-lacking region ("background") and plotted on the graph on the right panel. A total of 15 patches from 2 independent experiments were used to prepare right panels on (D) and (E). For statistical analysis, the unpaired $t$ test was used. ****$p < 0.0001$; ns, not significant. Scale bars, 1 μm. Data points that were used to create graphs are reported in S2 Table. F-actin, filamentous actin; pel, pellet; sup, supernatant; WT, wild-type.

and F-actin (14.9 ± 1.6 μM) was comparable to the previous study (13.2 ± 0.7 μM) [27]. However, the E16R Lifeact variant showed a 2-fold increased affinity for F-actin (6.0 ± 0.7 μM) compared to WT Lifeact–mCherry (Fig 3B and 3C). Although we could not observe density for E17 of Lifeact in our density map, we also created and tested an E17K variant of Lifeact, which should similarly create an additional interaction with E167 of actin and thereby increase the affinity of the peptide. In line with the prediction, the E17K Lifeact–mCherry variant also showed an increased affinity for F-actin ($K_d$ = 8.1 ± 0.6 μM) (Fig 3B and 3C). Together, these modifications show that based on our atomic model, Lifeact can be optimized by mutations to increase its binding to F-actin.

In the next step, we expressed the new Lifeact variants in yeast to test whether the E16R or E17K mutations improve Lifeact staining of F-actin also in vivo. In the confocal microscope, we observed that the Lifeact variants stained actin-rich structures in a similar way as WT Lifeact (Fig 2A and 2B). To compare Lifeact variants quantitatively, we performed 2 series of

analysis. Firstly, we calculated the ratio of Lifeact–mCherry to phalloidin fluorescence signal in patches (Fig 3D) in order to determine if the increased affinity of Lifeact to F-actin indeed increased amount of the fusion protein localized in the F-actin-rich patch. Secondly, to estimate the signal-to-noise ratio, we calculated the ratio of the Lifeact fluorescence signal in the patch to the Lifeact signal in the middle of the cell where no phalloidin staining was observed (Fig 3E). In both analyses, we saw only an insignificant increase of the average signal ratios of the new variants (Fig 3D and 3E). A reason for not seeing a stronger difference could be the high affinity of the probe to G-actin, which masks F-actin staining [21]. A follow-up study that tackles this issue should result in developing a probe with a better signal-to-noise ratio.

## Lifeact competes with cofilin and myosin

According to several studies, Lifeact staining interferes with cofilin binding to actin. For instance, Lifeact does not bind to cofilin-bound F-actin in cells [25], and Lifeact-expressing cells possess longer and thicker stress fibers. Studies by Flores and colleagues [28] suggested that reduced cofilin binding to F-actin is the underlying cause of the observed Lifeact-induced artefacts. Similarly, Lifeact causes changes in endocytosis and cytokinesis of *Schizosaccharomyces pombe*, which were attributed to reduced cofilin interaction with actin [27]. The authors of that study proposed that cofilin and Lifeact bind to different regions of F-actin and suggested that binding of one of these proteins impairs binding of the other by provoking a conformational change in F-actin [27].

Apart from Lifeact-induced stabilization of the closed conformation of the D-loop, however, our structure does not show major differences to previously reported structures of F-actin (S3B Fig) [7,8]. Therefore, a conformational change in F-actin cannot be the cause for the effect of Lifeact. When comparing our F-actin–Lifeact structure with that of F-actin–cofilin [31], it becomes obvious that the binding site of cofilin overlaps with that of Lifeact (Fig 4A). Notably, the same is true for myosin, which interacts with the same position on the actin surface (Fig 4A) [32]. We therefore performed in vitro competition actin-binding assays with human cofilin-1, the motor domain of human non-muscle myosin 2C (NM2C) isoform, and Lifeact. Lifeact successfully decreased cofilin and myosin binding in a dose-dependent manner (Fig 4B and 4C). As a negative control, we performed a similar competition assay with tropomyosin that binds to a different region of actin (Fig 4A) [33] and the Lifeact F10A mutant, which only binds weakly to F-actin (Fig 4B and 4C). As expected, the addition of Lifeact did not affect tropomyosin binding to F-actin nor could F10A Lifeact compete with cofilin-1 or NM2C (Fig 4B and 4C). Based on our structural and functional data, we demonstrate that the morphological artefacts described for Lifeact are not due to a conformational change in actin but are caused by competition for the same binding site on F-actin of Lifeact with actin-binding proteins, such as cofilin and myosin.

## Lifeact impairs the activity of bacterial toxins

In our laboratory, we study 2 bacterial toxins that interact with F-actin. One is *Pseudomonas aeruginosa* ExoY, a toxin that becomes a potent nucleotidyl cyclase upon interaction with F-actin [34]. After activation, the toxin generates a supraphysiologic amount of 3′,5′-cyclic guanosine monophosphate (cGMP) and 3′,5′-cyclic adenosine monophosphate (cAMP) that impedes cell signaling. The mutagenesis of D25 in actin abolishes ExoY binding to F-actin [35]. Since the same actin mutation also prevents Lifeact binding, we hypothesize that Lifeact and ExoY have overlapping binding sites.

The second toxin is the 30- kDa carboxyl terminus fragment of *P. luminescens* TccC3 (TccC3HVR), which is the effector domain of the large *Photorhabdus* toxin complex PTC3.

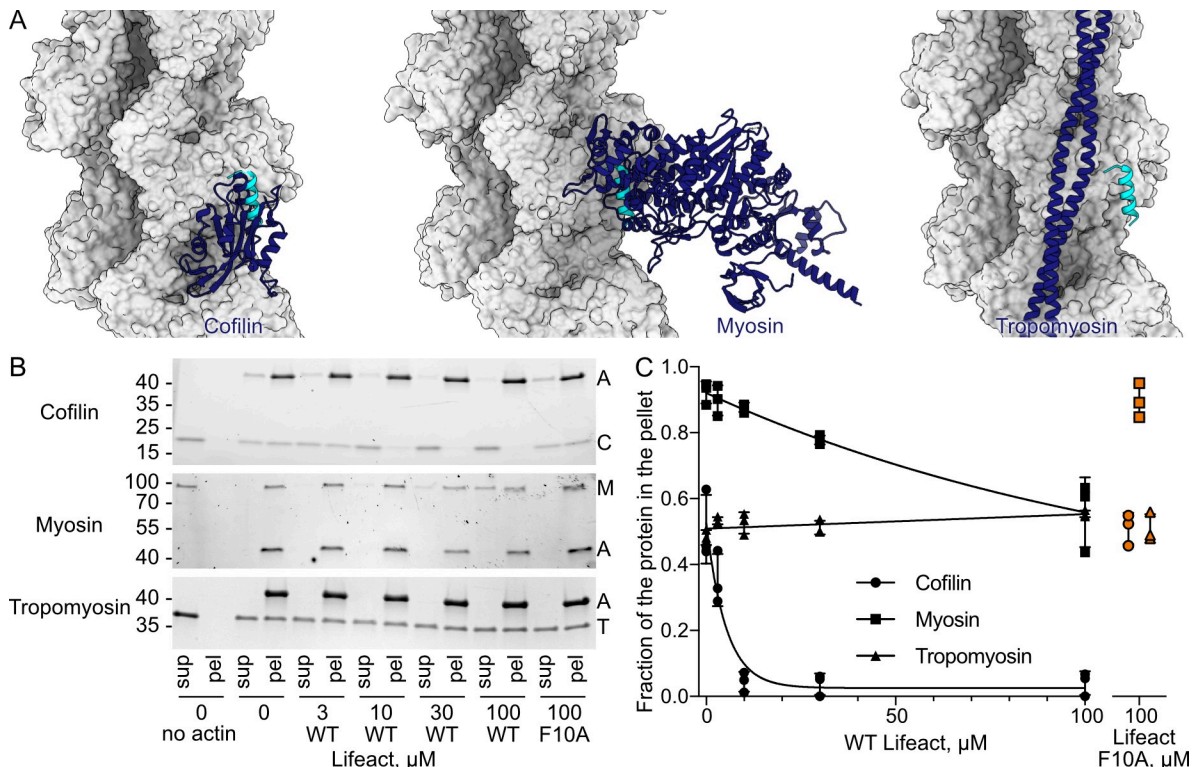

**Fig 4. Lifeact competes with cofilin and myosin in vitro.** (A) Structural models of the cofilin–F-actin (PDB 5YU8) [31], myosin–F-actin (PDB 5JLH) [32], and tropomyosin–F-actin (PDB 3J8A) [33] complexes. (B) SDS-PAGE analysis of cosedimentation experiments of F-actin (4 μM, upper band) with human cofilin-1 (4 μM, lower band); of F-actin (0.2 μM, lower band) with the motor domain of human NM2C (0.2 μM, upper band); and of F-actin (3 μM, lower band) with human tropomyosin alpha-1 (3 μM, upper band) in the presence of the indicated amounts of Lifeact. Representative gels are shown. The uncropped gels can be found in S4 Fig. The fractions of cofilin, myosin, and tropomyosin that cosedimented with F-actin in the corresponding experiments were quantified by densitometry and plotted against Lifeact concentrations at (C). The error bars correspond to standard deviations of 3 independent experiments. Data points that were used to create graphs are reported in S2 Table. A, F-actin; C, cofilin; F-actin, filamentous actin; M, myosin; pel, pellet; sup, supernatant; T, tropomyosin; NM2C, non-muscle myosin 2.

Once it is translocated into the cell by the injection machinery of PTC3, TccC3HVR acts as an ADP-ribosyltransferase that modifies actin at T148 [36]. This leads to uncontrolled actin polymerization, clustering, and finally, to cell death due to cytoskeletal collapse. T148 is located in close proximity to the Lifeact binding site (Fig 5C), therefore the actin binding site of TccC3HVR and Lifeact might overlap.

To understand whether Lifeact competes with the binding of ExoY, we first performed a cosedimentation assay with ExoY, F-actin, and different concentrations of Lifeact. In agreement with our hypothesis, we observed a decrease of ExoY binding to F-actin in the presence of Lifeact, whereas the Lifeact F10A mutant did not impair formation of the ExoY-F-actin complex (Fig 5A and 5B).

We then ADP-ribosylated F-actin by TccC3HVR in the presence of Lifeact. In our experimental setup, 3 μM of WT Lifeact was already sufficient to decrease the level of ADP ribosylation by a factor of 2, whereas in the control reaction, 100 μM of F10A Lifeact did not decrease the level of ADP-ribosylation at all (Fig 5D and 5E). This experiment strongly supports the hypothesis that TccC3HVR and Lifeact bind to the same region of F-actin.

Encouraged by these in vitro results, we decided to test whether expressing Lifeact in mammalian cells would protect them from the TccC3HVR toxin. We therefore expressed either

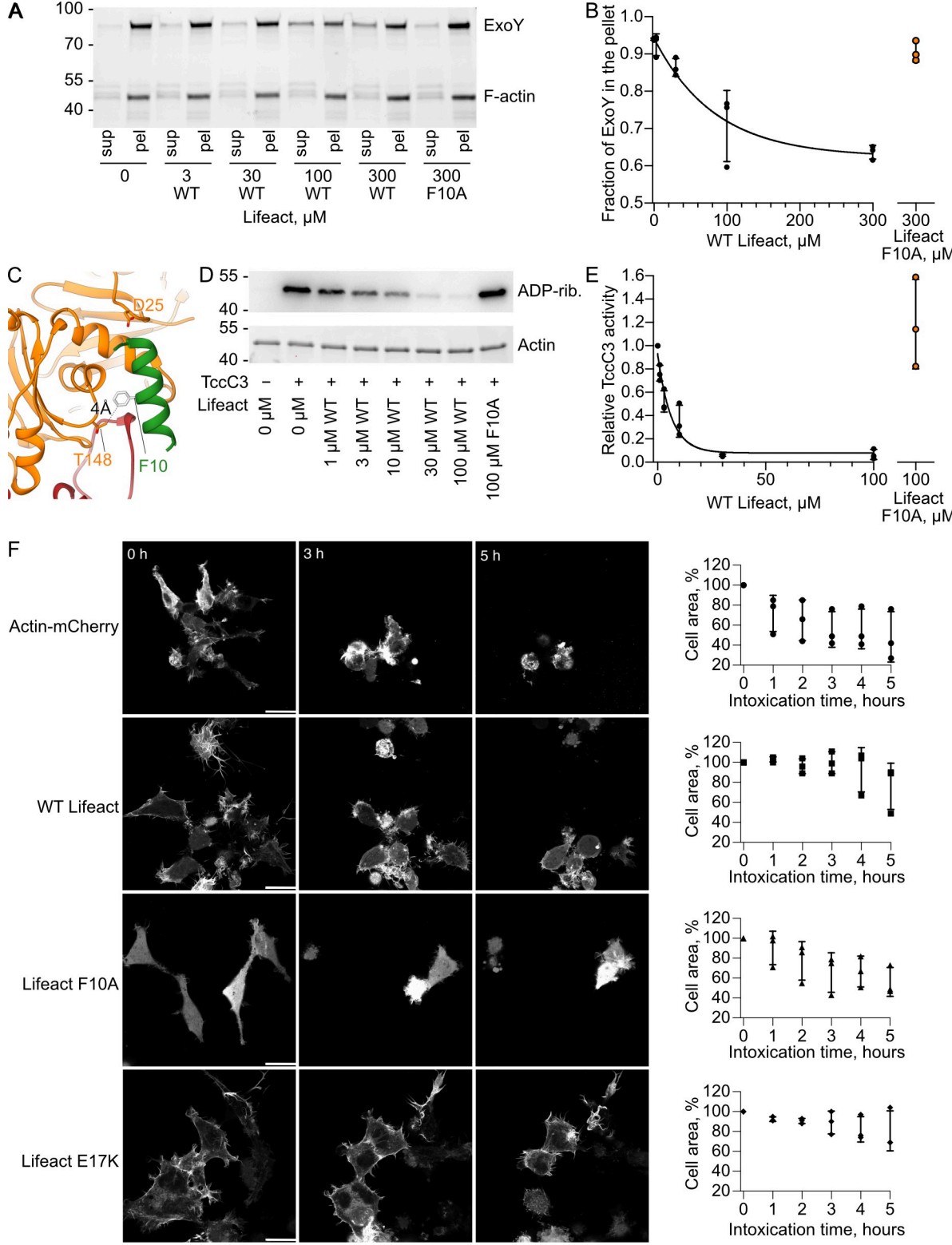

**Fig 5. Lifeact impairs the activity of F-actin-binding bacterial toxins.** (A) SDS-PAGE analysis of cosedimentation experiments of F-actin (1 μM, lower band) with ExoY-MBP (1 μM, upper band) in the presence of the indicated amounts of Lifeact. A representative gel is shown. The fractions of ExoY that cosedimented with F-actin were quantified by densitometry and plotted against Lifeact concentrations in (B). (C) The atomic model of the Lifeact–F-actin complex shows that T148, the site of Tcc3HVR modification [36], is localized within a 4 Å distance

from F10 of Lifeact (green). (D) The level of actin ADP-ribosylation by TccC3HVR in the presence of Lifeact was analyzed by western blot using an ADP-ribose binding reagent. The equal loading of actin was additionally verified by imaging the same stain-free gel prior to blotting (lower image). The ADP-ribosylation level of actin was quantified by densitometry and plotted against Lifeact concentrations in (E). Error bars at (B) and (E) correspond to standard deviations of 3 independent experiments. (F) HEK 293T cells expressing mCherry fusions of actin or LifeAct variants were intoxicated with 300 pM of the *Photorhabdus luminescens* toxin PTC3, which injects TccC3HVR into cells. The degree of cytoskeletal collapse and accompanying cell shrinkage was monitored for 5 hours using live cell imaging and is plotted based on 3 independent experiments for each condition. Scale bars, 20 μm. The uncropped western blots and gels can be found in S4 Fig. Data points that were used to create graphs are reported in S2 Table. F-actin, filamentous actin; HEK, human embryonic kidney; MBP, maltose-binding protein; pel, pellet; sup, supernatant; TccC3HVR, hypervariable region of TccC3; WT, wild-type.

mCherry-tagged WT Lifeact, Lifeact F10A, Lifeact E17K, or mCherry-tagged actin as a negative control in adherent HEK 293T cells. We then intoxicated the cells with PTC3 and observed the effect of the injected TccC3HVR. Our control cells expressing actin and cells expressing the F-actin-binding incompetent F10A Lifeact showed rapid cytoskeletal collapse and accompanying overall shrinkage (Fig 5F). However, the toxic effect of TccC3HVR was significantly reduced in cells that expressed WT Lifeact or the binding-competent E17K mutant. Thus, Lifeact has antitoxin properties and despite its effects on the cytoskeleton, it has the potential to be used as a precursor for the development of antitoxin drugs.

## Conclusions

In this study, we determined the binding site of Lifeact on F-actin and demonstrated that this peptide directly competes with actin-binding proteins such as cofilin and myosin, providing an explanation for how Lifeact alters cell morphology. In addition, we demonstrate how the affinity of Lifeact can be modulated by site-directed mutagenesis in order to create Lifeact-based probes with modified properties. Our results have strong implications for the usage of Lifeact as an actin filament label in fluorescent light microscopy and provide cell biologists with the background information that is needed to make a properly informed decision on whether to use Lifeact in an experiment. Furthermore, we have demonstrated that Lifeact competes with actin-binding toxins such as ExoY and TccC3HVR and partially counteracts the intoxication of cells by PTC3 toxin. This paves the way for the development of Lifeact-based antitoxin drugs.

## Supporting information

**S1 Table. Cryo-EM data collection, refinement, and validation statistics**
(DOCX)

**S2 Table. Excel file containing, in separate sheets, the numerical data for Figs 2B, 2D, 3C, 4C, 5B, 5E and 5F and S1E and S2A Figs.**
(XLSX)

**S3 Table. List of primers, strains, and plasmids used in this study.**
(DOCX)

**S1 Fig. Overview of the cryo-EM data.** (A, B) Example micrograph (A) and its power spectrum (B) at approximately −1.5-μm defocus. Filaments selected automatically by crYOLO are highlighted as differently colored dots. Scale bars, 10 μm. (C) Density map of the Lifeact–F-actin complex colored according to the local resolution. (D) Orientation distribution of the particles used in the final refinement round. (E) Fourier shell correlation (FSC) for the masked and unmasked final reconstructions. The FSC was calculated in the central 120 Å area of the map. Data points that were used to create this graph are reported in S2 Table.
(TIF)

**S2 Fig. Fit quality of the atomic model of Lifeact.** (A) Minimized energy of models of all possible registers of the Lifeact sequence into the density. Start1 and 2 correspond to the peptides starting at M1 or G2 with their N-termini pointing toward the pointed end of the filament. For Start1-rev and Start2-rev, the N-termini points toward the barbed end. The Rosetta energy values show a clear preferred solution. Data points that were used to create this graph are reported in S2 Table. (B) Density fit of the final model corresponding to the energy minimum seen in (A).
(TIF)

**S3 Fig. Comparison of the atomic models of F-actin and Lifeact–F-actin.** (A) The density maps and corresponding atomic models of Lifeact–F-actin–ADP–$P_i$–phalloidin in comparison to those of the open D-loop state in F-actin–ADP–$P_i$–phalloidin (PDB 6T1Y, EMDB 10363) [7] and closed D-loop state in F-actin–ADP–phalloidin (PDB 6T20, EMDB 10364) [7]. (B) Overall comparison of the atomic models of Lifeact–F-actin–ADP–$P_i$–phalloidin (red) and F-actin–ADP–$P_i$–phalloidin (beige, PDB 6T1Y, EMDB 10363) [7]. SD, subdomain.
(TIF)

**S4 Fig. Uncropped drop tests, western blots, and gels presented in the study.** *—This part of the gel was transferred onto PVDF membrane and stained with anti-MBP and anti-RPS9 serum. These western blots are presented on Fig 2E.
(TIF)

**S1 Movie. Stabilization of the D-loop in its closed state upon Lifeact binding.** (A) The density map of F-actin–ADP–$P_i$–phalloidin (EMDB 10363, gray) is superimposed with the corresponding atomic model (PDB 6T20) [7]. (B) Close-up view of the D-loop and the carboxyl terminus interface of F-actin–ADP–$P_i$–phalloidin complex. Possible position of Lifeact on F-actin–ADP–$P_i$–phalloidin shown in transparent green. Note that the D-loop is in its open state. (C) Close-up view on the interface between Lifeact and F-actin–ADP–$P_i$–phalloidin complex. I13 of Lifeact interacts with M47 of F-actin, stabilizing the closed D-loop conformation in F-actin. (D) Overview of the complete Lifeact–F-actin–ADP–$P_i$–phalloidin map and the corresponding atomic model.
(MP4)

**S1 File. Input protocols and starting files for designing Lifeact variants using Rosetta-Scripts.**
(ZIP)

## Acknowledgments

We thank S. Shydlovskyi for assistance with data collection, O. Hofnagel and D. Prumbaum for maintaining the EM facility, S. Bergbrede and M. Hülseweh for the excellent technical assistance, W. Linke and A. Unger (Ruhr-Universität Bochum, Germany) for providing us with muscle acetone powder, S. Rospert (University of Freiburg, Germany) for providing us anti-RPS9 serum, and S. Pospich and S. Maffini for fruitful discussions.

## Author Contributions

**Conceptualization:** Alexander Belyy, Stefan Raunser.

**Data curation:** Alexander Belyy, Felipe Merino, Oleg Sitsel.

**Formal analysis:** Alexander Belyy, Oleg Sitsel.

**Funding acquisition:** Stefan Raunser.

**Investigation:** Alexander Belyy, Felipe Merino, Oleg Sitsel, Stefan Raunser.

**Supervision:** Stefan Raunser.

**Validation:** Alexander Belyy, Felipe Merino, Oleg Sitsel, Stefan Raunser.

**Visualization:** Alexander Belyy, Felipe Merino, Oleg Sitsel.

**Writing – original draft:** Alexander Belyy, Felipe Merino, Oleg Sitsel.

**Writing – review & editing:** Stefan Raunser.

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
