## [Editor Report · Decision Letter 0]

26 Feb 2020

Dear Dr Raunser, 

Thank you for submitting your manuscript entitled "Structure of the Lifeact–F-actin complex" for consideration as a Research Article by PLOS Biology.

Your manuscript has now been evaluated by the PLOS Biology editorial staff as well as by an academic editor with relevant expertise and I am writing to let you know that we would like to send your submission out for external peer review.

Please re-submit your manuscript within two working days, i.e. by Feb 28 2020 11:59PM.

Kind regards,

Ines

--

Ines Alvarez-Garcia, PhD

Senior Editor

PLOS Biology

Carlyle House, Carlyle Road

Cambridge, CB4 3DN

+44 1223–442810

---

## [Decision Letter · Decision Letter 1]

28 Apr 2020

Dear Dr Raunser,

Thank you very much for submitting your manuscript "Structure of the Lifeact–F-actin complex" for consideration as a Research Article at PLOS Biology. Thank you also for your patience as we completed our editorial process, and please accept my apologies for the delay in providing you with our decision. Your manuscript has been evaluated by the PLOS Biology editors, an Academic Editor with relevant expertise, and by three independent reviewers.

As you will see, the reviewers find the conclusions interesting and significant for the field, however they also raise several issues. While the concerns brought up by Reviewers 2 and 3 are relatively minor, you would need to address Reviewer 1’s concerns. Although we won’t make essential for publication performing the experiments using time-lapse microscopy (Rev. 1, Point 3), testing the behaviour of the two Lifeact variants would be beneficial.

In light of the reviews (attached below), we will not be able to accept the current version of the manuscript, but we would welcome re-submission of a much-revised version that takes into account the reviewers' comments. We cannot make any decision about publication until we have seen the revised manuscript and your response to the reviewers' comments. Your revised manuscript is also likely to be sent for further evaluation by the reviewers.

We expect to receive your revised manuscript within 3 months. 

**IMPORTANT - SUBMITTING YOUR REVISION**

*Re-submission Checklist*

*Published Peer Review*

*PLOS Data Policy*

*Blot and Gel Data Policy*

Sincerely,

Ines

--

Ines Alvarez-Garcia, PhD

Senior Editor

PLOS Biology

Carlyle House, Carlyle Road

Cambridge, CB4 3DN

+44 1223–442810

Reviewers’ comments

Rev. 1:

In this manuscript, Belyy et al described a cryo-EM structure of Lifact bound actin filament and characterized the likely competition between Lifeact and other actin binding proteins. Lifeact is a widely used probe for actin filaments in vivo. However it has been well-documented that this probe, at a high intracellular concentration, perturbs cellular functions including endocytosis and cytokinesis. Although it has been proposed that Lifeact may interfere with the turnover of actin cytoskeletal structures, the molecular mechanism remained unclear, a question that this study tried to address.

Through this cryo-EM structure, Belyy et al. found that Lifeact, in a helical conformation, interacts with two neighboring actin monomers in a filament. Interestingly, Lifeact interacts specifically with the closed D-loop of one actin monomer. Lifeact competes directly against two essential actin binding proteins cofilin and myosin II to bind actin filaments. Finally, they demonstrated that Lifeact could be employed as a potential antagonist against two bacterial toxins ExoY and TccC3 which also bind actin.

Overall, the well-written manuscript provides novel insights into the interaction between Lifeact and actin filament both in vitro and in vivo. It will be of great interest to many cell biologists. However, the reviewer has several major concerns which are the following:

1) The reviewer is concerned about the novelty of this work. This will be the second cryo-EM structure of Lifeact-F-actin, after the first one was published about four months ago (Kumari et al. 2019 BioRxiv). Based on the manuscript, these two structures largely agree with each other, leading to many similar conclusions, even though the structure solved by Belyy et al. provides slightly higher resolution than the other one (3.5 vs. 4.2 angstrom). The reviewer would suggest that the authors focus more on describing and discussing the novel findings identified through their structure.

2) The biochemical studies of Lifeact and its mutants need substantial improvements. As an example, the reviewer would question why the authors used 300µM of Lifeact in many of their binding assays. Such high concentration of Lifeact may block nucleation of actin filaments because of its known interaction with actin monomer. As another example, the binding assay between actin filament and tropomyosin in the presence of Lifeact (Fig. 4H) was not well designed. Only one concentration (300µM) of Lifeact was used to compete against tropomyosin. More intermediate concentrations of Lifeact, as shown in Fig. 4C and 4E, shall be included to demonstrate that Lifeact doesn't block tropomyosin from binding to actin filaments.

3) The in vivo studies of Lifeact and its mutants are not very convincing. For example, the micrographs of actin cytoskeletal structures in budding yeast (Fig. 2A) are of poor quality. Of the three prominent actin structures in budding yeast, only the patch is visible even though the actin cables and the contractile ring shall have been identifiable as well. Another example is the quantification of the localization of Lifeact or its mutants in actin patches (Fig. 2B). It would have been much more convincing to examine this using time-lapse microscopy of the actin patches. Through this, it would be possible to quantify the localization of Lifeact mutants in the patch. Lastly, the authors failed to provide any evidence that the two novel Lifeact variants (Fig. 3) are better probes than the wild-type Lifeact. A in vivo experiment that uses these two variants E16R and E17K to stain actin cytoskeletal structures could easily test this hypothesis.

4) Some key experiments lack proper quantification. Most importantly, the binding affinity between Lifeact/Lifeact mutants and actin filaments shall have been quantified (Fig.3C-D) and compared to many previous studies (Riedle 2008 and Courtemanche 2016). In another example, no quantitative evidence is provided to support the claim that Lifeact doesn't modify the structure of actin filament. A more detailed comparison between the structure of Lifeact bound actin filament and that of unbound actin filament (Oda et al. 2009), will be highly informative.

Minor concerns:

1) Fig. 1: It would be important to provide an overview on which subdomains of the actin monomer, from SD1 to SD4, interact with Lifeact.

2) Fig. 2C: It is not clear from the text whether the endogenous actin had been replaced with the two actin mutants (D25Y and L349M), an important condition to interpret this result.

3) Fig. 3D: The binding between the actin filament and wild-type Lifeact never reached a plateau, which makes it hard to estimate the affinity between them. Higher concentrations of Lifeact shall be used to saturate the filaments.

4) Fig. 5A-B: The representative gel picture is less than ideal. A smudge obscured part of the lane where the supernatant for 300 µM Lifeact was. Please replace it.

5) Fig. 5F: The plot is difficult to understand. It is highly preferable to replace it with time courses of average value ± standard deviation at each data point.

6) Please scale back the statement that Lifeact compete with the bacterial toxins to bind actin filament. The study provided no evidence that Lifeact competes directly with either of the bacterial toxins to bind actin filaments. The inhibitory effect of Lifeact to Tcc3HVR and ExoY in vivo could be indirect.

7) Please clarify which myosin the authors referred to throughout the main text and figure legends. The reviewer assumed that it is non-muscle myosin II.

In summary, this study by Belyy et al. has the potential to make a significant contribution to our understanding of Lifeact but it requires major revisions. The reviewer will be happy to review the revised manuscript when it becomes available.

Rev. 2:

Summary

This is a timely and important paper that describes new observations on a widely used actin-binding probe molecule, including a high-resolution co-complex with actin filaments. Key observations from the structure include the prediction that Lifeact binding interferes with binding of key cytoskeletal cofactors including myosin and cofilin, as well as bacterial toxins. These predictions are borne out by biochemical studies and in vivo studies where they show that Lifeact competes with myosin and cofilin and interferes with bacterial toxin activity.They further modify Lifeact properties with structure-guided design, supported by binding measurements. The structure presented here also provides additional evidence to support prior predictions by this group that actin's key 'D-loop' element can adopt discrete 'open' and 'closed' conformations, whose equilibria are potentially affected by ATP hydrolysis and cofactor binding .

Overall Sentiment

This paper is well written and succinct. I have no reservations to publish it immediately. Understanding the molecular basis of Lifeact-F-actin binding will enable researchers to use it with greater prudence and sophistication.

Major concerns

None

Minor concerns

The two sentences of the results/discussion are a little off, although the intended meaning can be gotten from them. I think the message that is intended is that (1) prior studies have indicated a molecular basis for stabilization of actin filaments by phalloidin (2) prior studies indicate phalloidin places actin in a well defined nucleotide state with a well-defined D-loop conformation, making it a suitable reference state for structure studies with Lifeact.

Rev. 3:

Lifeact is a chemically modified 17-mer peptide derived from the N-terminal end of the methyltransferase ABP140, an actin-binding protein able to mediate actin localization. It is today one of the most versatile live-cell actin tools with which to visualize actin dynamics in the cell. Genetically encoded actin reporters, either fluorescent derivatives of actin or actin-binding domains, although powerful tools for studying actin filament architecture and dynamics in live cells, are biased probes and their cellular distribution does not accurately reflect that of the cytoskeleton. The differences in live-cell actin probe localization can be associated to efficiency of incorporation capacity, competition with endogenous actin associated proteins, and differential rates of actin filament turnover. In addition to their fidelity with which they reveal the architecture of the actin cytoskeleton, special attention to establishing expression levels that do not alter cell morphology or behavior is required.

Biochemical and structural studies defining the mechanisms underlying the actin binding interactions of actin probes could become powerful tools, providing mechanistic insight into the formation of actin networks in live cells. Here, Raunser and colleagues address this exact point, and thus this work is of importance.

The team presents a 3.5-Å structure of Lifeact / phalloidin-stabilized F-actin complex using single particle cryo-EM. In the reconstruction densities corresponding to ADP, Mg2+ and Pi in the nucleotide-binding pocket of actin and a density corresponding to phalloidin at the expected positions were all identify. Of the Lifeact peptide, 16 out of 17 amino acids were unambiguously fit into the density. Lifeact interacts with a hydrophobic binding pocket on F-actin and stretches over two adjacent actin subunits, stabilizing the DNase I binding loop of actin in the closed conformation. Through actin co-sedimentation assays the authors showed Lifeact competes with cofilin (assuming no F-actin change) and myosin, and suggest that the morphological artefacts described for Lifeact are caused by competition for the same binding site on F-actin. In addition, Lifeact competes with actin-binding toxins such as ExoY and TccC3HVR, demonstrating the potential of Lifeact to serve as a platform for anti-toxin drugs. The team also employed site directed mutagenesis to provide Lifeact probes with modified properties expanding the functional role of this probe.

One point that requires a bit more rigorous testing: The relevance of the co-sed assays that requires over 100umol of Lifeact to show effect seems a bit puzzling in terms of its relevance to in vivo conditions. One possible solution would be to perform a fluorescence quenching experiment with pre-assembled Lifeact F-actin. Adding myosin or cofilin should show quenching behavior hopefully at less unphysiological concentrations.

---

## [Decision Letter · Decision Letter 2]

19 Aug 2020

Dear Stefan,

Thank you for submitting your revised Research Article entitled "Structure of the Lifeact–F-actin complex" for publication in PLOS Biology. Please accept again my apologies for the delay in sending you our decision, but are currently very few editors handling a big number of manuscripts. I have now obtained advice from two of the original reviewers and have discussed their comments with the Academic Editor. 

We're delighted to let you know that we're now editorially satisfied with your manuscript. However before we can formally accept your paper and consider it "in press", we also need to ensure that your article conforms to our guidelines. A member of our team will be in touch shortly with a set of requests. As we can't proceed until these requirements are met, your swift response will help prevent delays to publication. Please also make sure to address the data and other policy-related requests noted at the end of this email.

*Copyediting*

*Published Peer Review History*

*Early Version*

*Submitting Your Revision*

Sincerely,

Ines

--

Ines Alvarez-Garcia, PhD,

Senior Editor,

ialvarez-garcia@plos.org,

PLOS Biology

DATA POLICY: IMPORTANT, PLEASE READ

Fig. 2B, D; Fig. 3C, D, E; Fig. 4C; Fig. 5B, E, F and Fig. S2A

Reviewers’ comments

Rev. 1:

In this revised manuscript, Belyy et al has addressed all the major concerns raised in my previous review. Overall, the authors have significantly improved the quality of both the biochemical studies of Lifeact and the microscopy data. The reviewer particularly applauds the effort by the authors to measure the affinity between Lifeact or its variants and actin filaments with a substantially improved binding assay (Fig. 3C). Importantly, the revised study also determined whether the E16R and E17K variants can be employed as improved actin filament markers (Fig. 2A). Although neither has proved to be a better probe compared to the original Lifeact for now, the reviewer finds the result very informative. The authors are on the right track to potentially optimize this widely-used actin marker.

The review has no more concern and would suggest acceptance of this revised manuscript.

Rev. 3:

The revisions, modifications and additional set of experiments addressed the review concerns adequately.

---

## [Editor Report · Decision Letter 3]

26 Oct 2020

Dear Dr Raunser,

On behalf of my colleagues and the Academic Editor, Carole A Parent, I am pleased to inform you that we will be delighted to publish your Research Article in PLOS Biology. 

PRODUCTION PROCESS

Before publication you will see the copyedited word document (within 5 business days) and a PDF proof shortly after that. The copyeditor will be in touch shortly before sending you the copyedited Word document. We will make some revisions at copyediting stage to conform to our general style, and for clarification. When you receive this version you should check and revise it very carefully, including figures, tables, references, and supporting information, because corrections at the next stage (proofs) will be strictly limited to (1) errors in author names or affiliations, (2) errors of scientific fact that would cause misunderstandings to readers, and (3) printer's (introduced) errors. Please return the copyedited file within 2 business days in order to ensure timely delivery of the PDF proof. 

If you are likely to be away when either this document or the proof is sent, please ensure we have contact information of a second person, as we will need you to respond quickly at each point. Given the disruptions resulting from the ongoing COVID-19 pandemic, there may be delays in the production process. We apologise in advance for any inconvenience caused and will do our best to minimize impact as far as possible.

EARLY VERSION

PRESS 

Kind regards,

Alice Musson

Publishing Editor, 

PLOS Biology

on behalf of

Ines Alvarez-Garcia,

Senior Editor

PLOS Biology